# Analysis of the Curative Effect of Temporomandibular Joint Disc Release and Fixation Combined with Chitosan Injection in the Treatment of Temporomandibular Joint Osteoarthrosis

**DOI:** 10.3390/jcm12041657

**Published:** 2023-02-19

**Authors:** Chuan-Bin Wu, Hai-Jiang Sun, Ning-Ning Sun, Qing Zhou

**Affiliations:** Department of Oral and Maxillofacial Surgery, China Medical University, Shenyang 110122, China

**Keywords:** temporomandibular joint, osteoarthrosis, chitosan, VAS, mouth opening, disc

## Abstract

Objective: Temporomandibular joint osteoarthritis (TMJ-OA) is common in clinic. The purpose of this study was to evaluate the efficacy of disc release, fixation and chitosan injection in the treatment of TMJ-OA. Methods: From March 2021 to March 2022, 32 patients who underwent the unilateral reduction and fixation of temporomandibular joint disc release were retrospectively studied. All patients were diagnosed with TMJ-OA and were treated with chitosan injection. This group of patients was analyzed by the visual analog scale (VAS) for pain and improvement of maximum comfortable mouth opening before treatment and 6 months after treatment. A paired t-test was used to evaluate the treatment effect, and *p* < 0.05 indicated that the difference was statistically significant. Results: All 32 patients were successfully treated by surgery and chitosan injection in the second week after operation. The duration of disease in this group ranged from 1 to 10 months, with an average of 5.7 months. After 6 months of follow up, 30 patients were satisfied with the treatment and two were unsatisfied. The difference in the treatment effect was found to be statistically significant (*p* < 0.05). Conclusions: Temporomandibular joint disc release and fixation combined with chitosan injection is effective in the treatment of TMJ-OA.

## 1. Introduction

Temporomandibular joint osteoarthrosis (TMJOA) is a chronic degenerative disease occurring locally in the TMJ and is an organic disease of the joint, often resulting in the degeneration and loss of articular cartilage, accompanied by pathological changes in subchondral bone and other soft and hard tissues [1,2]. Clinical symptoms include pain in the joint area or muscle groups around the joint, various joint murmurs, limited mouth opening, and mandibular dyskinesia [3,4]. Temporomandibular joint disorders (TMD) can be divided into muscle disorders (including myofascial pain with and without mouth opening limitation) and intra-articular disorders (including disc displacement with or without reduction and mouth opening limitation, arthralgia, and arthritis) [5]. TMD is the second most common musculoskeletal disorder that causes pain and disability [6,7]. Although nonsurgical interventions, such as physical and medical treatments, are more prevalent in the clinic, surgical treatment, including open and arthroscopic procedures, is required in some TMJOA patients with severe symptoms [8,9,10]. However, in long term follow up studies, there have been differences in the time and effect of postoperative functional recovery in patients with TMJOA, and postoperative adjuvant therapy has affected the prognosis to some extent.

Chitosan is a safe and reliable polysaccharide polymer with excellent biocompatibility, degradability and biological activity. It is often used to prevent tissue adhesions after surgery and has been used in the conservative treatment of osteoarthropathy [11,12]. Chitosan’s mechanisms of action are as follows: it inhibits the proliferation of fibroblasts, reduces scar hyperplasia after operation, and avoids the occurrence of adhesion; as a viscoelastic supplement, it reduces the fluidity of joint synovial fluid to help lubricate joints; based on its macromolecular three-dimensional network structure, it can form a physical barrier, protect articular cartilage, reduce tissue friction, and help to reduce postoperative pain; it promotes the synthesis of cartilage extracellular matrix to regulate the metabolism of chondrocytes; and based on its ability to cross the cartilage surface, it acts as a mechanical barrier that inhibits the release of inflammatory factors, which relieves pain [13]. Therefore, this study will evaluate the therapeutic effect of chitosan injection in patients with TMJOA undergoing temporomandibular joint disc reduction and fixation. We hypothesize that surgery combined with postoperative chitosan injection is effective in the treatment of TMJOA.

## 2. Materials and Methods

All subjects gave their informed consent for inclusion before they participated in the study. The study was conducted in accordance with the Declaration of Helsinki, and the protocol was approved by the Ethics Committee of the Stomatological Hospital Affiliated with the China Medical University (201914).

Informed Consent Statement: Informed consent was obtained from all subjects involved in the study.

A retrospective study was conducted. We looked to the past to determine whether the surgery combined with chitosan injection was effective in the treatment of TMJOA.

### 2.1. Subjects

Thirty two TMJOA patients who underwent unilateral reduction and fixation of temporomandibular joint disc release from March 2021 to March 2022 in the Department of Oral and Maxillofacial Surgery of the Affiliated Dental Hospital of China Medical University were analyzed retrospectively. According to Wikes-Bronstein stages, temporomandibular disorders are divided into five stages. Phase I and II involve the reducible anterior displacement of the articular disc, and phase II has more pain symptoms than phase I. Phase III, IV and V involve the irreducible anterior displacement of the articular disc. Condylar bone destruction occurs in phase IV, and perforation of the articular disc takes place in phase V [14]. All patients were asked to have a temporomandibular joint MRI examination before surgery.

This study used the following inclusion criteria: (1) the same surgeon performed both the joint disc reduction and fixation; (2) mouth opening training was conducted after the operation; and (3) there was good compliance and follow-up within the prescribed time. The exclusion criteria were as follows: (1) loss to follow up or follow up time < 6 months; (2) a history of temporomandibular joint surgery; and (3) other diseases or mental diseases affecting joint and jaw development.

These 32 patients were asked to receive a chitosan injection two weeks after operation. All patients were analyzed by visual analog scale (VAS) for pain and improvement of maximum comfortable mouth opening before treatment and 6 months after treatment.

### 2.2. Therapies

All patients underwent the reduction and fixation of temporomandibular joint disc release. The operation method was as follows: After the subcutaneous injection of a 1:100,000 epinephrine saline solution into the affected side, the standard preauricular incision was made and bluntly separated to the superficial layer of deep temporal fascia from this plane down to the level of the zygomatic arch (Figure 1). The facial nerve was further dissected downward and outside the joint capsule of the temporomandibular ligament, and the layer where the facial nerve was located was pushed to the superficial layer for protection.

A “T”-shaped incision was performed outside the joint capsule to open it. The joint disc was found at the posterior slope of the articular eminence and was attached before electrocoagulation separation and cutting. The joint anchor nail was implanted 0.5 cm below the posterior slope of the condylar process to cover the joint disc at the top of the condylar process. The joint disc was then fixed with the anchor nail (Figure 2). Bipolar electrocoagulation was sufficient to stop bleeding, and a 100 mL negative pressure drainage tube was placed in the layer-by-layer suture wound (Figure 3). The postoperative routine included the use of antibiotics and steroids to prevent infection and reduce swelling. Antibiotics: Cefuroxime sodium 1500 mg + 0.9% saline solution 250 mL, intravenous drip, bid, three days. Steroids: Dexamethasone 10 mg + 5% glucose injection 250 mL, intravenous drip, qd, three days. In addition, physiotherapy of two weeks in duration was suggested.

Upon follow-up 2 weeks after surgery, treatment with intra-articular injection of chitosan was performed by the same oral and maxillofacial surgeon. Chitosan was injected into the articular cavity (Chittier, Shanghai Qisheng Biologics Co., Ltd., Shanghai, China) using the following method: The patient was placed in a supine position with the head deviated to the healthy side, with the needle point 8–10 mm away from the front of the ear screen on the line connecting the middle point of the ear screen and the outer canthus. The operation area was disinfected. A No. 5 needle was used to puncture the skin along the inlet needle, and the superior joint cavity was entered approximately 45° forward, upward and inward. After no blood was drawn back, 1 mL of a mixture of vitamin B12 and 1% lidocaine was given. With the needle in place, the syringe was replaced, and 1 mL of chitosan was injected (Figure 4). After the injection was completed, the patient was observed for 5–10 min and was able to leave following the absence of any adverse reactions.

### 2.3. Evaluation Index

Pain scores and maximum comfortable mouth opening were recorded before treatment and 6 months after treatment in each group. Patients were scored for pain using the visual analog scale (VAS). This method uses a scale of 0 to 10 cm that represents the degree of clinical symptoms: 0 indicates no clinical symptoms, and 10 indicates unbearable clinical symptoms [2,15]. The patient’s active maximum mouth opening was measured [16,17] with a straight edge.

### 2.4. Statistical Processing

The data were statistically analyzed using SPSS 20.0 software. A paired t-test was used to compare the improvement in pain VAS scores and maximal comfort mouth opening before and six months after treatment. A *p* < 0.05 indicated that the difference was statistically significant.

## 3. Results

Thirty-two TMJ-OA patients aged 32–75 years (average 51.4 ± 13.7) were recruited in the study. No patients had complications such as ankylosis and salivary fistula, after the operation; however, six patients(18.8%) had temporary facial paralysis, mainly manifested as difficulty in raising the eyebrows and incomplete eyelid closure; and they were given oral neurotrophic drugs and all they all had recovered within the 6-month follow up. Other postoperative adverse reactions included swelling, pain, and numbness in the operative area. The maximum preoperative VAS score was 8, and the minimum was 5, with a mean of 6.4 ± 0.8; the maximum postoperative VAS score was 7, and the minimum was 2, with a mean of 3.8 ± 1.1. The preoperative maximum comfortable mouth opening was 36 mm, the minimum was 19 mm, and the mean was 26.8 ± 5.1 mm. The postoperative maximum comfortable mouth opening was 39 mm, the minimum was 29 mm, and the mean was 35.0 ± 2.0 mm. The duration of disease in this group ranged from 1 to 10 months, with an average of 5.7 ± 2.9 months. Thirty patients were satisfied with the treatment and two were unsatisfied, and the difference in the treatment effect was *p* < 0.05, indicating that the difference was statistically significant (Table 1).

Some patients may complain about the expense of getting an MRI or the discomfort experienced with getting an MRI. A total of 10 patients underwent post-operative MRI. New bone formation was found in five patients (Appendix A).

## 4. Discussion

In this study, the mean of preoperative VAS is 6.4 ± 0.8; the mean of postoperative VAS was 3.8 ± 1.1. The mean of preoperative maximum comfortable mouth was 26.8 ± 5.1 mm; the mean of the postoperative maximum comfortable mouth opening was 35.0 ± 2.0 mm. All patients received a chitosan injection on the affected side in the second postoperative week. After 6 months of follow-up, 93.8% of the patients were satisfied with the treatment, indicating its effectiveness. In some cases, new bone was found.

The occurrence of TMJOA has a long history, and studies have shown that conservative interventions, such as physical ones, have some positive effects on alleviating pain and improving joint movement and function, but surgical treatment [18,19] is recommended for patients with more severe TMJOA, especially those with obvious anterior disc displacement. The reduction and fixation of the temporomandibular joint disc may cause damage to the skin, muscle, temporal and zygomatic branch of the facial nerve, and the superficial and middle temporal arteries and veins of the auricle and temple, resulting in local scarring, auricle and temple tissue collapse and eyebrow-raising dyskinesia. However, surgery also significantly relieves pain in the joint area, improves the maximal mouth opening, and significantly mitigates the adverse effects of joint symptoms on daily life.

Hyaluronic acid (HA) is a natural component of normal synovial fluid. HAs mechanical role in lubrication, its multiple biological effects, and its metabolic function as a nutrient component has made the intra-articular injection of HA an important clinical method for the treatment of TMJOA [20]. HA not only plays a viscous role in mechanically reducing friction, but also regulates the proteolytic enzyme activity of plasminogen activators and prevents the indirect activation of inflammatory mediators. In addition, nitric oxides (NOs) are free radicals involved in neurotransmission and vasodilation, and HAs reduce NO production and joint pain [21]. A systematic review of the clinical efficacy of intra-articular injections of HA [22] indicated that for TMJOA, there was no significant difference in pain reduction and functional recovery between intra-articular injections of HA or joint punctures with or without HA injections.

It has also been reported in the literature that corticosteroids are used for superior articular injection in the treatment of TMJOA. Corticosteroids relieve TMJOA symptoms by reducing inflammation associated with arthritis-related changes. In addition, studies have shown that glucocorticoids deposit in the joint cavity after injection, causing the destruction of cartilage and subchondral bone [23,24].

It has been demonstrated in animal experiments that chitosan has sustained-release absorption properties, is effective in maintaining local drug concentrations, increases the duration and intensity of drug use and is superior to other drugs in intra-articular injections [25]. The physicochemical properties of chitosan are similar to those of intra-articular aminopolysaccharide, which develop the foundation of cartilage and the cartilage matrix formation and metabolism and promote the repair and regeneration of damaged articular cartilage [26]. Based on this, it can be concluded that chitosan treatment may have a positive effect on the growth of condylar cartilage after TMJOA.

Antimicrobials are often required during and after oral and maxillofacial surgery, but the incidence of resistance events is increasing as antimicrobials are promoted. Spectral antimicrobials often lead to changes in the patient’s own bacterial counts, resulting in an imbalance of normal flora and the emergence of new drug-resistant strains. As a result this, we pay attention to sterilization before surgery and keep antibiotic treatment as short as possible [27,28].

The study has shortcomings. In this study, age, sex and duration are variables that may be effective with regard to the final results. Thus, a statistical analysis was conducted. There is no statistical significance concerning age and sex. However, the short duration group showed better improvement of maximum comfortable mouth opening (Figure 5). In future studies, we will exclude these distractions. Furthermore, 32 patients is a relatively small sample, and we suggest that more patients be recruited with sufficient pre- and post-imaging materials. Thirdly, the study has no control group. However, we used a paired t-test to assess the therapeutic effect. The paired t-test, a self-controlled statistical method that can eliminate interference associated with outcome variables, was applied.

## 5. Conclusions

The study focused its attention on the therapeutic effect of surgery combined with a chitosan injection. In our cohort, each patient received the same treatment. Given the follow up results, we conclude that the approach was effective. Appropriate adjuvant therapy (i.e., chitosan intra-articular injection) can significantly improve the function of the TMJ in patients with TMJOA after reduction, TMJ disc release, and fixation, and it is recommended to popularize its use in the clinic.

## Figures and Tables

**Figure 1 jcm-12-01657-f001:**
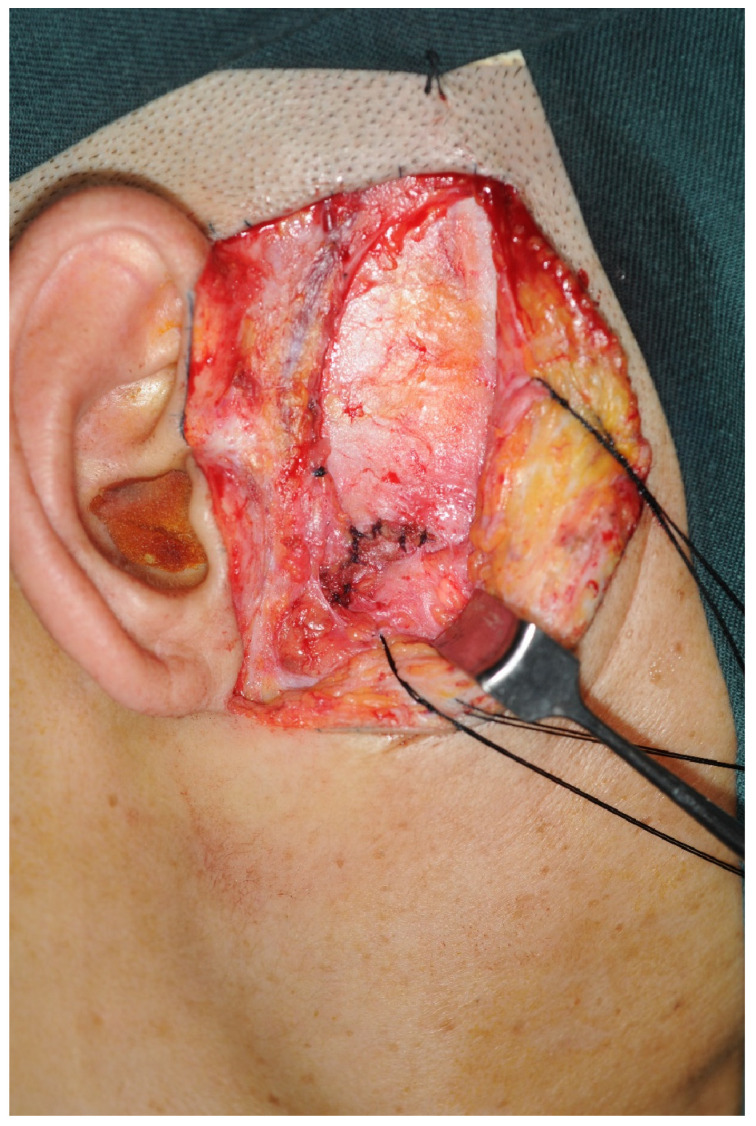
The superficial layer of deep temporal fascia was revealed.

**Figure 2 jcm-12-01657-f002:**
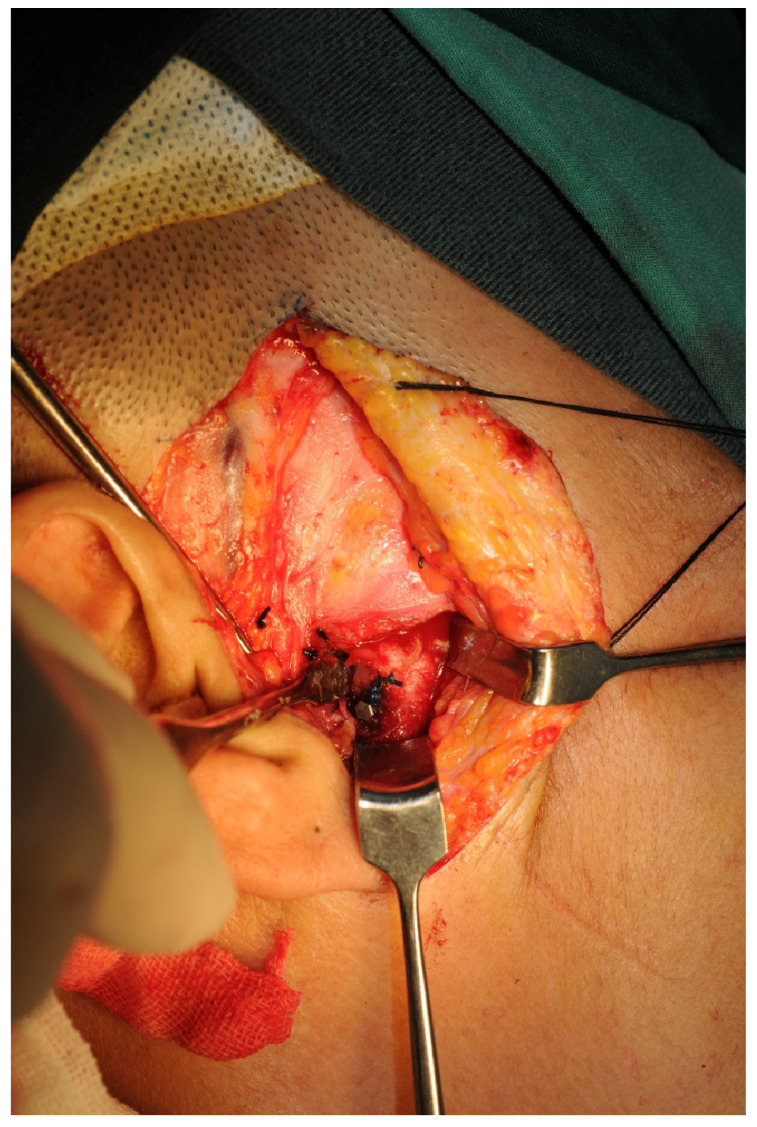
The anterior disc displacement was released and fixed on an anchor.

**Figure 3 jcm-12-01657-f003:**
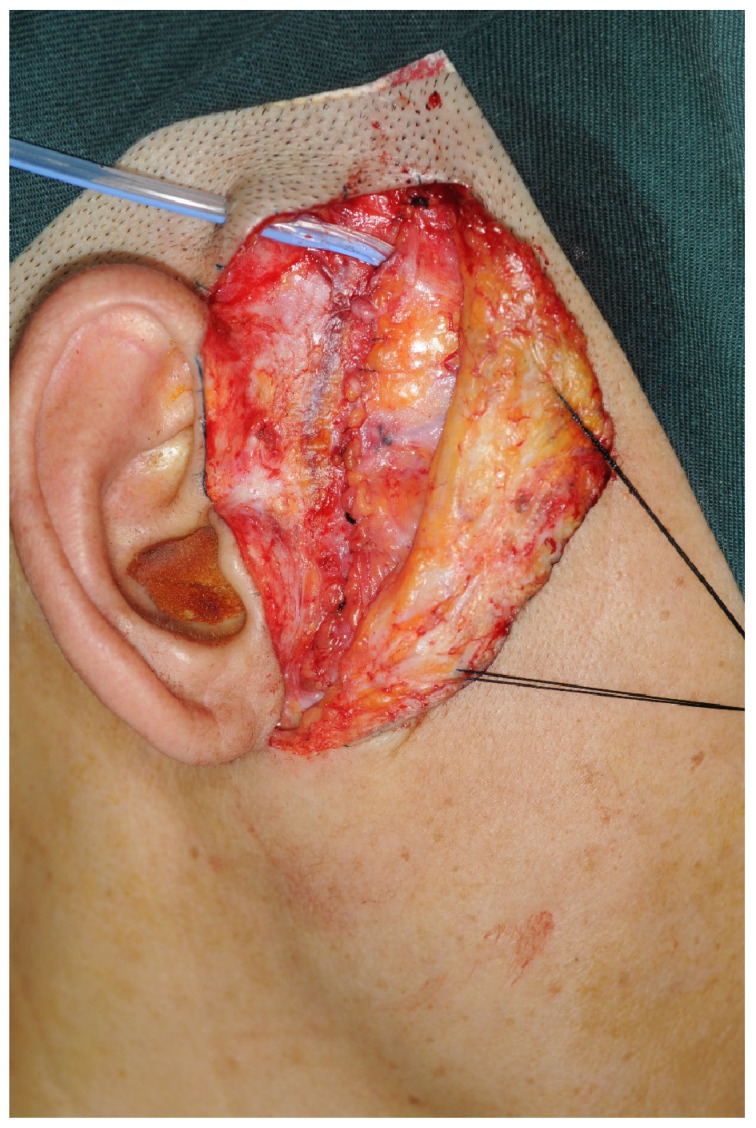
A 100 mL negative pressure drainage tube was placed under the deep temporal fascia.

**Figure 4 jcm-12-01657-f004:**
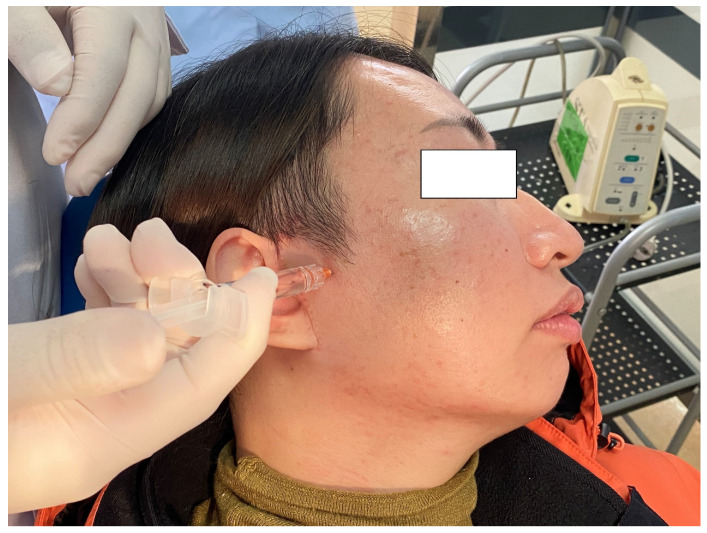
Chitosan injection was performed.

**Figure 5 jcm-12-01657-f005:**
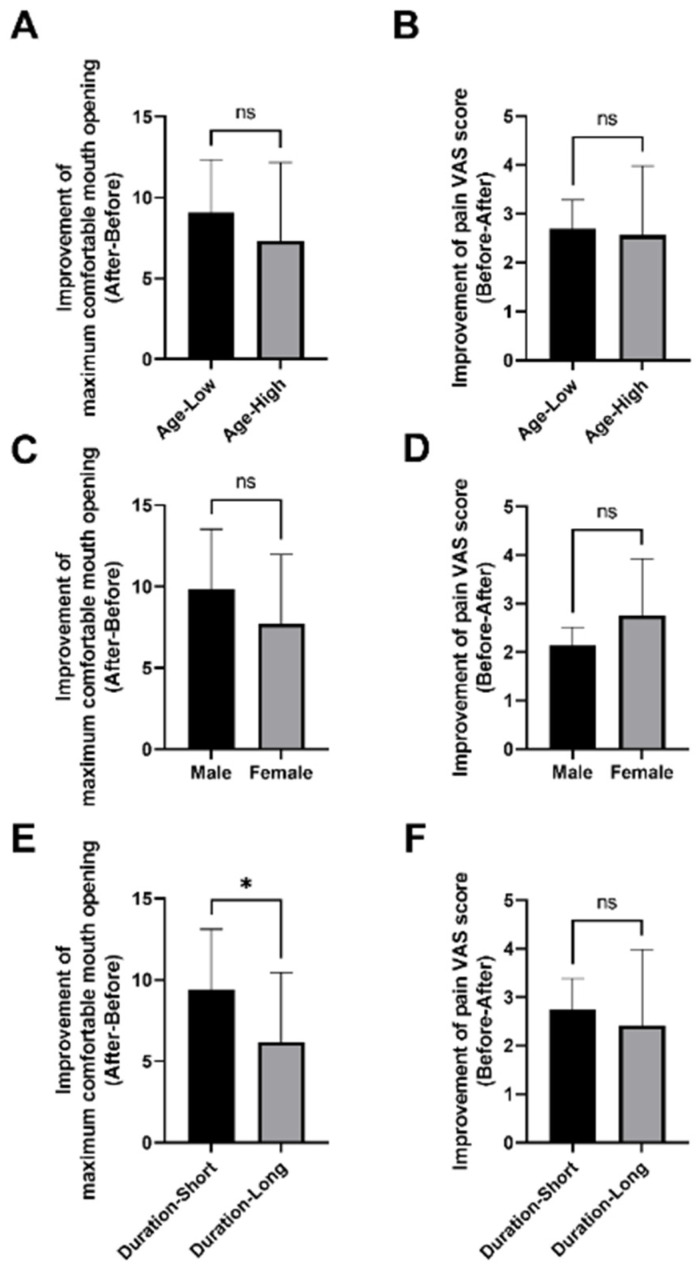
Statistical results of age, sex and duration. ns: no significance; *: *p <* 0.05. (**A**) Improvement of maximum comfortable mouth opening; (**B**) Improvement of pain VAS score; (**C**) Improvement of maximum comfortable mouth opening; (**D**) Improvement of pain VAS score; (**E**) Improvement of maximum comfortable mouth opening; (**F**) Improvement of pain VAS score.

**Table 1 jcm-12-01657-t001:** Patient characteristics.

No.	Age	Gender	Duration (Month)	Affected Sides	Intervention	Satisfied	Maximum Comfortable Mouth Opening (mm)		Pain VAS	
							Before	After	Before	After
1	45	Male	2	L	Surgery + Injection	Yes	20	36	6	4
2	62	Female	4	R	Surgery + Injection	Yes	19	35	7	4
3	70	Female	6	R	Surgery + Injection	Yes	20	35	8	4
4	33	Female	7	L	Surgery + Injection	Yes	23	35	6	3
5	32	Male	2	B	Surgery + Injection	Yes	24	32	6	4
6	71	Female	8	R	Surgery + Injection	No	35	35	7	7
7	60	Female	10	L	Surgery + Injection	Yes	35	37	6	2
8	51	Female	2	L	Surgery + Injection	Yes	35	39	5	3
9	39	Female	5	B	Surgery + Injection	Yes	30	35	6	3
10	46	Female	6	R	Surgery + Injection	Yes	21	35	6	4
11	42	Male	8	R	Surgery + Injection	Yes	29	37	5	3
12	63	Male	3	L	Surgery + Injection	Yes	23	32	6	4
13	41	Female	6	B	Surgery + Injection	Yes	19	29	7	4
14	49	Female	5	R	Surgery + Injection	Yes	22	35	7	3
15	69	Female	4	R	Surgery + Injection	Yes	26	34	8	5
16	58	Male	10	B	Surgery + Injection	Yes	20	34	5	3
17	39	Female	5	L	Surgery + Injection	Yes	30	36	7	4
18	75	Female	7	L	Surgery + Injection	No	36	36	6	7
19	72	Female	9	B	Surgery + Injection	Yes	30	36	7	3
20	39	Female	2	L	Surgery + Injection	Yes	25	35	6	3
21	34	Female	10	R	Surgery + Injection	Yes	26	35	8	5
22	42	Male	2	B	Surgery + Injection	Yes	27	35	7	5
23	36	Female	1	B	Surgery + Injection	Yes	30	36	7	4
24	30	Female	6	R	Surgery + Injection	Yes	30	36	6	3
25	63	Female	12	L	Surgery + Injection	Yes	30	35	6	2
26	65	Female	6	R	Surgery + Injection	Yes	27	37	7	4
27	58	Female	7	B	Surgery + Injection	Yes	30	36	6	3
28	59	Female	5	L	Surgery + Injection	Yes	20	30	7	4
29	68	Male	8	R	Surgery + Injection	Yes	30	36	6	3
30	38	Female	4	R	Surgery + Injection	Yes	30	35	6	4
31	46	Female	2	R	Surgery + Injection	Yes	27	36	6	3
32	51	Female	8	L	Surgery + Injection	Yes	30	36	6	4

## Data Availability

No new data were created.

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
