# Peer review of "Analysis of the Curative Effect of Temporomandibular Joint Disc Release and Fixation Combined with Chitosan Injection in the Treatment of Temporomandibular Joint Osteoarthrosis"

_jcm, 2023, doi:10.3390/jcm12041657_

Round 1

Reviewer 1 Report (New Reviewer)

This study investigated evaluate the efficacy of disc release, fixation and chitosan injection in the treatment of temporomandibular joint osteoarthrosis. There are some major issue which should be addressed:

1. The type of study should be mentioned in both abstract and main text.

2. Kindly add a history of previous studies and the available information on this subject to the introduction. Please tell what is the point that differentiates your work in introduction part.

3. Kindly add null hypothesis to the introduction

4. "All patients were asked to receive chitosan injection two weeks later after operation. This group of patients was analyzed by visual analog scale (VAS)" it is not clear. All patients or group of patients received chitosan injection?

5. What test did you use for evaluation of normal distribution of data?

6. What was the distribution of patients in terms of sex frequency? The age or sex could be effective on the final results? It should be discussed in the discussion and also limitations.

7. Suggestions for the future study should be added

8. I did not find any comparisons of the obtained results to the other similar studies in discussion part. The discussion part is similar to the introduction.

Author Response

This study investigated evaluate the efficacy of disc release, fixation and chitosan injection in the treatment of temporomandibular joint osteoarthrosis. There are some major issue which should be addressed:

 Question 1:

1.The type of study should be mentioned in both abstract and main text.

Reply: Thanks for your question. This is a retrospective study and we mentioned in both abstract (From March 2021 to March 2022, 32 patients who underwent unilateral reduction and fixation of temporomandibular joint disc release were retrospectively studied) and Materials and Methods (A retrospective study was conducted. We look to the past to know whether the surgery combined with chitosan injection is effective in the treatment of TMJOA).

Question 2:

2.Kindly add a history of previous studies and the available information on this subject to the introduction. Please tell what is the point that differentiates your work in introduction part.

Reply: Thanks for your question. surgical treatment, including open and arthroscopic procedures, is required in some TMJOA patients with severe symptoms[8,9,10]. However, in long term follow up studies, there has been differences in the time and effect of postoperative functional recovery in patients with TMJOA, and postoperative adjuvant therapy has affected the prognosis to some extent.

Question 3:

  1. Kindly add null hypothesis to the introduction

Reply: Thanks for your question. We hypothesis that surgery combined with chitosan injection postoperative is effective in the treatment of TMJOA.

Question 4:

  1. "All patients were asked to receive chitosan injection two weeks later after operation. This group of patients was analyzed by visual analog scale (VAS)" it is not clear. All patients or group of patients received chitosan injection?

Reply: Thanks for your question. We re-wrote the sentence: These 32 patients were asked to receive chitosan injection two weeks later after operation. All patients were analyzed by visual analog scale (VAS) for pain and improvement of maximal mouth opening comfort before treatment and 6 months after treatment.

Question 5:

  1. What test did you use for evaluation of normal distribution of data?

Reply: Thanks for your question. We used One-way analysis of variance, Bonferroni.

Question 6:

  1. What was the distribution of patients in terms of sex frequency? The age or sex could be effective on the final results? It should be discussed in the discussion and also limitations.

Reply: Thanks for your question. In this study, age, sex and duration are variables that may be effective on the final results. So, statistical analysis was conducted. There is no statistical significance on age and sex. However, the short duration group showed better improvement of maximum comfortable mouth opening. Thus, the study has short comings. Next time, we had better exclude these distractions.

Question 7:

  1. Suggestions for the future study should be added

Reply: Thanks for your question. Besides, 32 patients is relative small and we suggest a lot more patients recruited with enough pre and post imaging materials.

Question 8:

  1. I did not find any comparisons of the obtained results to the other similar studies in discussion part. The discussion part is similar to the introduction.

Reply: Thanks for your question. In this study, age, sex and duration are variables that may be effective on the final results. So, statistical analysis was conducted. There is no statistical significance on age and sex. However, the short duration group showed better improvement of maximum comfortable mouth opening (Figure 5). All patients received chitosan injection on the affected side in the second postoperative week. After 6 months of follow up, 93.8% of the patients were satisfied with the treatment, indicating the effectiveness of the treatment. In some cases, new bone was found.

Reviewer 2 Report (New Reviewer)

In the work entitled "Analysis of the curative effect of temporomandibular joint disc release and fixation combined with chitosan injection in the treatment of temporomandibular joint osteoarthrosis"  the authors aimed to describe the differences in terms of pain and maximal mouth opening improvement of TMJ disc release and fixation with adjunctive chitosan injection.

Despite the overall reasonable quality of the manuscript, several corrections and additions are mandatory.

Line 69: the point n. 2 is not clear. Please provide a clear inclusion criteria.

Line 95: the authors cited antibiotics and steroids. Please provide a detailed drugs prescription for these two therapies.

Every time that a mean value appears (e.g. lines 127, 132, 133, 134, 136), please provide its standard deviation between parentheses. 

Table 1 must follow its citation in the text, after line 138. Its position at the end of the article, after the references, makes the understanding difficult.

The same must be done for images and graphics.

Please provide a caption for graphics.

In matherials and methods the authors cited an antibiotic therapy in the post-operative phase. In the discussion this aspect must be improved, since  in oral and maxillofacial surgery the use of antibiotics has been focused also by WHO. Please add necessarily these references in the discussion:

https://pubmed.ncbi.nlm.nih.gov/35065702/

https://pubmed.ncbi.nlm.nih.gov/32867465/

Author Response

Reviewer 2:

In the work entitled "Analysis of the curative effect of temporomandibular joint disc release and fixation combined with chitosan injection in the treatment of temporomandibular joint osteoarthrosis"  the authors aimed to describe the differences in terms of pain and maximal mouth opening improvement of TMJ disc release and fixation with adjunctive chitosan injection.

Despite the overall reasonable quality of the manuscript, several corrections and additions are mandatory.

Question 1.

Line 69: the point n. 2 is not clear. Please provide a clear inclusion criteria.

Reply: Thanks for your question. We re-wrote n.2: mouth opening training was conducted after operation.

Question 2.

Line 95: the authors cited antibiotics and steroids. Please provide a detailed drugs prescription for these two therapies.

Reply: Thanks for your question. We added the following part:

Antibiotics: Cefuroxime sodium 1500mg + 0.9% saline solution 250ml, intravenous drip, bid, three days. Steroids: Dexamethasone 10mg + 5% glucose injection 250ml, intravenous drip, qd, three days.

Question 3.

Every time that a mean value appears (e.g. lines 127, 132, 133, 134, 136), please provide its standard deviation between parentheses. 

Reply: Thanks for your question. We have made modifications as requested.

Question 4.

Table 1 must follow its citation in the text, after line 138. Its position at the end of the article, after the references, makes the understanding difficult.

Reply: Thanks for your question. We modified as required.

Question 5.

The same must be done for images and graphics.

Reply: Thanks for your question. We modified as required.

Question 6.

Please provide a caption for graphics.

Reply: Thanks for your question. We deleted the graphics and provided captions for images and tables.

Question 7.

In materials and methods the authors cited an antibiotic therapy in the post-operative phase. In the discussion this aspect must be improved, since in oral and maxillofacial surgery the use of antibiotics has been focused also by WHO. Please add necessarily these references in the discussion:

https://pubmed.ncbi.nlm.nih.gov/35065702/

https://pubmed.ncbi.nlm.nih.gov/32867465/

Reply: Thanks for your question. Antimicrobials are often required during and after oral and maxillofacial surgery, but the incidence of resistance events is increasing as antimicrobials are promoted. Spectral antimicrobials often lead to changes in the patient's own bacterial values, resulting in an imbalance of normal flora and the emergence of new drug-resistant strains. Based on this, we pay attention to sterilization before surgery and keep antibiotics as short as possible[27,28].

  1. Antimicrobial Resistance Collaborators. Global burden of bacterial antimicrobial resistance in 2019: a systematic analysis[J]. Lancet, 2022,399(10325):629-655.
  2. S D'Agostino, M Dolci. Antibiotic therapy in oral surgery: a cross sectional survey among Italian dentists[J]. J Biol Regul Homeost Agents, 2020,34(4):1549-1552.

Reviewer 3 Report (New Reviewer)

Dear authors and editor,

The manuscript titled "Analysis of the curative effect of temporomandibular joint disc release and fixation combined with chitosan injection in the treatment of temporomandibular joint osteoarthrosis" aims to evaluate the efficacy of discrelease, fixation and chitosan injection in the treatment of temporomandibular joint osteoarthrosis

There are many minor and major issues I'd like the authors resolve.

Abstract

1-Add the study design to the abstract. Also, the authors can choose to add the study design to the title of the manuscript.

2-Change the keywords.  Not found in the MeSH (Medical Subject Headings). 

3-It is recommended to add a background or introduction section to the abstract.

Introduction

4- Further information related to chitosan injection is recommended. Move the information from the section discussion. 

Materials and Methods

5-It is recommended to add a section that includes the design of the study. I recommend putting it at the beginning of the methodology next to the ethical considerations.

6- It is recommended that this sentence be moved to the "Statistical Processing" : "A paired t-test was used to evaluate the treatment effect, 76 and P<0.05 indicated that the difference was statistically significant".

7-Study size: Explain how the study size was arrived at.  The sample size is very important; I consider this to be a pilot study, a cross-sectional descriptive study.

Results

8-Measures of central tendency such as the mean must be accompanied by the standard deviation. "Thirty-two TMJ-OA patients aged 32-75 years (average 51.4) were recruited in the study. "

9-It is recommended to analyze if there was a difference between the phases of the patients evaluated. I consider that this is a variable to be taken into account, and it could also be evaluated if there are differences with respect to the duration of the disease.

Discussion

10-It is recommended to start the discussion with the most relevant findings found in the results of the study.

11-More suitable concepts are described for the introduction section.  "Chitosan is a safe and reliable polysaccharide polymer with excellent biocompatibility, degradability and 168 biological activity. It is often used to prevent tissue adhesions after surgery and has been used in the conserva- 169 tive treatment of osteoarthropathy [22-23]. Chitosan’s mechanisms of action are as follows: it inhibits  the proliferation of fibroblasts, reduces scar hyperplasia after operation and avoids the occurrence of adhesion;  as a viscoelastic supplement, it reduces the fluidity of joint synovial fluid to help lubricate joints; based on its  macromolecular three-dimensional network structure, it can form a physical barrier, protect articular cartilage,  reduce tissue friction, and help to reduce postoperative pain; it promotes the synthesis of cartilage extracellular  matrix to regulate the metabolism of chondrocytes; and based on its ability to cross the cartilage surface, it  acts as a mechanical barrier that inhibits the release of inflammatory factors which relieves pain [24]."

12-It is recommended that figures 5-6 be moved to a supplementary material section.

Conclusión

13-It is recommended to create a section called conclusions

Referencia:

14-It is recommended to follow the journal's guidelines for bibliography.

Author Response

Dear authors and editor,

The manuscript titled "Analysis of the curative effect of temporomandibular joint disc release and fixation combined with chitosan injection in the treatment of temporomandibular joint osteoarthrosis" aims to evaluate the efficacy of discrelease, fixation and chitosan injection in the treatment of temporomandibular joint osteoarthrosis

There are many minor and major issues I'd like the authors resolve.

Abstract

Question 1:

  • Add the study design to the abstract. Also, the authors can choose to add the study design to the title of the manuscript.

Reply: Thanks for your question. From March 2021 to March 2022, 32 patients who underwent unilateral reduction and fixation of temporomandibular joint disc release were retrospectively studied.

Question 2:

  • Change the keywords.  Not found in the MeSH (Medical Subject Headings). 

Reply: Thanks for your question. We changed the keywords: Temporomandibular joint; Osteoarthrosis; Chitosan; VAS; Mouth opening; Disc

Question 3:

  • It is recommended to add a background or introduction section to the abstract.

Reply: Thanks for your question. Because of the limitation word counts of the abstract, we added one sentence as background: “Temporomandibular joint osteoarthritis (TMJ-OA) is common in clinic, ”

Introduction

Question 4:

  • Further information related to chitosan injection is recommended. Move the information from the section discussion. 

Reply: Thanks for your question. We moved the following part from the section discussion:  Chitosan is a safe and reliable polysaccharide polymer with excellent biocompatibility, degradability and biological activity. It is often used to prevent tissue adhesions after surgery and has been used in the conservative treatment of osteoarthropathy [11-12]. Chitosan’s mechanisms of action are as follows: it inhibits the proliferation of fibroblasts, reduces scar hyperplasia after operation and avoids the occurrence of adhesion; as a viscoelastic supplement, it reduces the fluidity of joint synovial fluid to help lubricate joints; based on its macromolecular three-dimensional network structure, it can form a physical barrier, protect articular cartilage, reduce tissue friction, and help to reduce postoperative pain; it promotes the synthesis of cartilage extracellular matrix to regulate the metabolism of chondrocytes; and based on its ability to cross the cartilage surface, it acts as a mechanical barrier that inhibits the release of inflammatory factors which relieves pain [13]. Therefore, this study will evaluate the therapeutic effect of chitosan injection in patients with TMJOA undergoing temporomandibular joint disc reduction and fixation.

Materials and Methods

Question 5:

5-It is recommended to add a section that includes the design of the study. I recommend putting it at the beginning of the methodology next to the ethical considerations.

Reply: Thanks for your question. We added a section as you recommended: A retrospective study was conducted. We look to the past to know whether the surgery combined with chitosan injection is effective in the treatment of TMJOA.

Question 6:

6- It is recommended that this sentence be moved to the "Statistical Processing" : "A paired t-test was used to evaluate the treatment effect, 76 and P<0.05 indicated that the difference was statistically significant".

Reply: Thanks for your question. We moved the sentence to the "Statistical Processing": A paired t-test was used to compare the improvement in pain VAS scores and maximal comfort mouth opening before and six months after treatment. A P<0.05 indicated that the difference was statistically significant.

Question 7:

7-Study size: Explain how the study size was arrived at.  The sample size is very important; I consider this to be a pilot study, a cross-sectional descriptive study.

Reply: Thanks for your question. Given the inclusion and exclusion criteria, 32 patients were recruited from March 2021 to March 2022. These patients underwent unilateral reduction and fixation of temporomandibular joint disc release. On the other hand, they got chitosan injection 2 weeks later after surgery.

Results

Question 8:

8-Measures of central tendency such as the mean must be accompanied by the standard deviation. "Thirty-two TMJ-OA patients aged 32-75 years (average 51.4) were recruited in the study. "

Reply: Thanks for your question. We modified as required.

Question 9:

9-It is recommended to analyze if there was a difference between the phases of the patients evaluated. I consider that this is a variable to be taken into account, and it could also be evaluated if there are differences with respect to the duration of the disease.

Reply: Thanks for your question. We added the following part:

The study has short comings. In this study, age, sex and duration are variables that may be effective on the final results. So, statistical analysis was conducted. There is no statistical significance on age and sex. However, the short duration group showed better improvement of maximum comfortable mouth opening (Figure 5).

Discussion

Question 10:

10-It is recommended to start the discussion with the most relevant findings found in the results of the study.

Reply: Thanks for your question. We started the discussion with the results of the study.

In this study, the mean of preoperative VAS is 6.4±0.8; the mean of postoperative VAS was 3.8±1.1. The mean of preoperative maximum comfortable mouth was 26.8±5.1mm; the mean of postoperative maximum comfortable mouth opening was 35.0±2.0 mm. All patients received chitosan injection on the affected side in the second postoperative week. After 6 months of follow up, 93.8% of the patients were satisfied with the treatment, indicating the effectiveness of the treatment. In some cases, new bone was found.

Question 11:

11-More suitable concepts are described for the introduction section.  "Chitosan is a safe and reliable polysaccharide polymer with excellent biocompatibility, degradability and 168 biological activity. It is often used to prevent tissue adhesions after surgery and has been used in the conserva- 169 tive treatment of osteoarthropathy [22-23]. Chitosan’s mechanisms of action are as follows: it inhibits  the proliferation of fibroblasts, reduces scar hyperplasia after operation and avoids the occurrence of adhesion;  as a viscoelastic supplement, it reduces the fluidity of joint synovial fluid to help lubricate joints; based on its  macromolecular three-dimensional network structure, it can form a physical barrier, protect articular cartilage,  reduce tissue friction, and help to reduce postoperative pain; it promotes the synthesis of cartilage extracellular  matrix to regulate the metabolism of chondrocytes; and based on its ability to cross the cartilage surface, it  acts as a mechanical barrier that inhibits the release of inflammatory factors which relieves pain [24]."

Reply: Thanks for your question. We moved these parts to the introduction section.

Question 12:

12-It is recommended that figures 5-6 be moved to a supplementary material section.

Reply: Thanks for your question. We modified as requested.

Conclusión

Question 13:

13-It is recommended to create a section called conclusions

Reply: Thanks for your question. We added conclusion part:

  1. Conclusions

The study focus its attention on the therapeutic effect of surgery combined with chitosan injection. In our cohort, each patient received the same approach. Given the follow up results, we deem that the approach is effective. Appropriate adjuvant therapy (i.e., chitosan intra-articular injection) can significantly improve the function of the TMJ in patients with TMJOA after reduction, TMJ disc release and fixation, and it is recommended to popularize its use in the clinic.

Referencia:

Question 14:

14-It is recommended to follow the journal's guidelines for bibliography.

Reply: Many thanks for your suggestions. We unified the reference.

Round 2

Reviewer 1 Report (New Reviewer)

This manuscript has been modified according to the reviewer's comments. Thank you for your effort.

Reviewer 3 Report (New Reviewer)

Thank you very much for your response. The authors have answered all the considerations raised. It is true that the sample size is small. This lowers the power of such a study. I recommend the authors to continue with the line of research.

This manuscript is a resubmission of an earlier submission. The following is a list of the peer review reports and author responses from that submission.

Round 1

Reviewer 1 Report

The paper looks like a clinical trial for using chitosan injection after disc reduction and fixation

No ethical approval was mentioned

No control group is included

The study doesn’t follow any guidelines like surgical classification criteria for TMJ BY

Dimilroilis 2013 or Wilkes classification 2013

The criteria for follow-up were based on mouth opening and analog scale, although two MRIs were shown in the discussion section, which should be mentioned in the method and material section.

The effect of surgery or injection of chitosan is effective in reliving patient discomfort can not be concluded from this study, a controlled trial should be conducted

Author Response

Dear Editors and Reviewers:

RE: Analysis of the curative effect of temporomandibular joint disc release and fixation combined with chitosan injection in the treatment of temporomandibular joint osteoarthrosis (ID: jcm-2071099)

We would like to thank the Journal of clinical medicine for giving us the opportunity to revise our manuscript. Your positive and constructive comments and suggestions are all valuable and helpful for revising and improving our paper.

We provide this document to explain, point by point, the details of our revisions in the manuscript and our responses to your comments as follows.
We are looking forward to hearing from you soon.

Kind Regards,

Yours sincerely,

Qing Zhou

E-mail: cbwqz8060@163.com

Analysis of the curative effect of temporomandibular joint disc release and fixation combined with 2 chitosan injection in the treatment of temporomandibular joint osteoarthrosis

Review #1: The paper looks like a clinical trial for using chitosan injection after disc reduction and fixation

Question 1.

No ethical approval was mentioned

Reply:Thanks for your question. We added the ethical approval.  All subjects gave their informed consent for inclusion before they participated in the study. The study was conducted in accordance with the Declaration of Helsinki, and the protocol was approved by the Ethics Committee of Stomatological Hospital Affiliated to China Medical University (201914).

Informed Consent Statement: Informed consent was obtained from all subjects involved in the study.

The corresponding part is on page 2 with red color (line 47-line 50).

Question 2. No control group is included.

Reply: Thanks for your question. The study has shortcoming because of no control group. But, we used paired t test to assess the therapeutic effect. As we know, the paired t test, a self-controlled statistical method that can eliminate interference associated with outcome variables, was applied. This study focus its attention on the treatment effect of surgery combined with chitosan injection. The corresponding part is on page 9 with red color (line 215 to line 220).

Question 3. The study don’t follow any guidline like surgical classification criteria fot TMJ BY Dimilroilis 2013 or Wilkes classification 2013

Reply: Thanks for your question. We agree with you. According to Wikes-Bronstein stages, temporomandibular disorders are divided into 5 stages (I. e. I-V stage).Phase I and II belong to reducible anterior displacement of articular disc, and phase II has more pain symptoms than phase I. Phase III, IV and V belong to irreducible anterior displacement of articular disc. Condylar bone destruction occurs in phase IV, and perforation of articular disc occurs in phase V. All patients were asked to have temporomandibular joint MRI examination before surgery and they were in phase III—V. The corresponding part is on page 2 with red color (line 54 to line 59).

Question 4. The criteria for follow up was based on mouth opening and analogue scale although two MRI were shown in discussion section should be mentioned in method and material section.

Reply: Thanks for your question. All patients were asked to have temporomandibular joint MRI examination before surgery and they were in phase III—V. The corresponding part is on page 2 with red color (line 58 to line 59).

Question 5. The effect of surgery or injection of chitosan is effective in reliving patient discomfort can not be concluded from this study, a control trial should be conducted

Reply: Thanks for your question. The study focus its attention on the therapeutic effect of surgery combined with chitosan injection. In our cohort, each patient received the same approach. Given the follow up results, we deem that the approach is effective. As to the control trial, we tried to reply you on Question 2. The corresponding part is on page 9 with red color (line 218 to line 220).

We appreciate your work earnestly, and hope that the correction will meet with approval. Once again, thank you very much for your comments and suggestions.

Reviewer 2 Report

i compliment with the authors for the excellent results and for the nice intraoperative images.

the paper is interesting but in my humble opinion needs to be substantially improved

INTRODUCTION . line 45 . reference 5 is not completely adherent. other references should be added , specifically concerning the different reported surgical results achieved with different surgical methods. this topic is very controversial  and the introduction should better describe the actual state of the art.

MATERIALS AND METHODS.  the diagnosis and staging of TMJOA is very important. the authors should precise which classification or staging method they adopt. TMJOA has different levels of severity which atre very important for the results. I imagine all the patient underwent MRI and/or CT. I would like to have more informations about diagnosis , staging and indications for surgical treatment of TMJOA. Inclusion criteria should comprehend these informations.

line 67: the anterior approach looks like an anterior incision in the pre auricular skin crease. This approach it is not very popular in europe  where standard pre auricular or  endoauricular variants are more used : would the authors mind to describe more clearly the cutaneous incision.

line 77 : what is the maxillary nodule? would you mind to use a different more common anatomical term?

line 84 : hormones ? is it steroids?  the physiotherapy protocol should be better described .

RESULTS : authors should precise the incidence and duration of temporary facial nerve defect if they mention it (line119) : " some "is no scientific information.

the authors should precise at what stage of follow up the reported VAS and opening values were registered.

did  the patient undergo post op MRI ? what are the results in terms of stability of disk repositioning ? was the new bone formation reported in image 5 and 6 common in all the patients ? 

it would be very interesting to know the result of post op imaging.

concerning image 5 and  6 they should be anonimyzed : it is possible  to read the name of the patient.

IMAGE 5-6  there is also  problem with the date: the PREOP MRI seems performed in 2019; the post op MRI is performed on 4/2019 according to the date reported in the scan . How is this possible?

 in images5-6 the age reported on the scan for the patient is 13 years old ( born 4/21/2006).  the material and method section report ages of the patient included in the 32-75 years range. How is that possible ?

DISCUSSION : the overall structure of the discussion should be reformulated . clear references should be made to the literature reported results for TMJOA , according to stages and different techniques.

lines 144-147  should be reformulated . which are the difference inputs op functional recovery if the surgical methods is always the same ?

the conclusions . especially concerning occlusion relationships and reconstruction of internal structure of the joint  are not supported by the data reported in the study  (lines204-210). in  my opinion the paragraph should be reformulated and made more adherent to the data reported in the study.

Author Response

Dear Editors and Reviewers:

RE: Analysis of the curative effect of temporomandibular joint disc release and fixation combined with chitosan injection in the treatment of temporomandibular joint osteoarthrosis (ID: jcm-2071099)

We would like to thank the Journal of clinical medicine for giving us the opportunity to revise our manuscript. Your positive and constructive comments and suggestions are all valuable and helpful for revising and improving our paper.

We provide this document to explain, point by point, the details of our revisions in the manuscript and our responses to your comments as follows.
We are looking forward to hearing from you soon.

Kind Regards,

Yours sincerely,

Qing Zhou

E-mail: cbwqz8060@163.com

Review #2:

i compliment with the authors for the excellent results and for the nice intraoperative images.

the paper is interesting but in my humble opinion needs to be substantially improved

Question 1.

INTRODUCTION . line 45 . reference 5 is not completely adherent. other references should be added , specifically concerning the different reported surgical results achieved with different surgical methods. this topic is very controversial  and the introduction should better describe the actual state of the art.

Reply:Thanks for your question. We added other references and re-wrote the part.

Question 2.

MATERIALS AND METHODS.  the diagnosis and staging of TMJOA is very important. the authors should precise which classification or staging method they adopt. TMJOA has different levels of severity which atre very important for the results. I imagine all the patient underwent MRI and/or CT. I would like to have more informations about diagnosis , staging and indications for surgical treatment of TMJOA. Inclusion criteria should comprehend these informations.

Reply:Thanks for your question. According to Wikes-Bronstein stages, temporomandibular disorders are divided into 5 stages (I. e. I-V stage).Phase I and II belong to reducible anterior displacement of articular disc, and phase II has more pain symptoms than phase I. Phase III, IV and V belong to irreducible anterior displacement of articular disc. Condylar bone destruction occurs in phase IV, and perforation of articular disc occurs in phase V. All patients were asked to have temporomandibular joint MRI examination before surgery and they were in phase III—V. The corresponding part is on page 2 with red color (line 54 to line 59).

Question 3.

line 67: the anterior approach looks like an anterior incision in the pre auricular skin crease. This approach it is not very popular in europe  where standard pre auricular or  endoauricular variants are more used : would the authors mind to describe more clearly the cutaneous incision.

Reply:Thanks for your question. After the subcutaneous injection of a 1:100000 epinephrine saline solution into the affected side, the standard pre auricular incision was done The corresponding part is on page 2 with red color (line 74 to line 76).

Question 4.

line 77 : what is the maxillary nodule? would you mind to use a different more common anatomical term?

Reply:Thanks for your question. We use articular eminence instead. The corresponding part is on page 3 with red color (line 85 to line 86).

Question 5.

line 84 : hormones ? is it steroids?  the physiotherapy protocol should be better described .

Reply:Thanks for your question. The postoperative routine included the use of antibiotics and steroids to prevent infection and reduce swelling. Besides, physiotherapy was suggested for two weeks. The corresponding part is on page 3 with red color (line 91 to line 93).

Question 6.

RESULTS : authors should precise the incidence and duration of temporary facial nerve defect if they mention it (line119) : " some "is no scientific information.

Reply:Thanks for your question. We agree with you. however, six patients(18.8%) had temporary facial paralysis, mainly manifested as difficulty in raising eyebrows and incomplete eyelid closure, and they were given oral neurotrophic drugs and all the six patients recovered with 6month follow up. The corresponding part is on page 6 with red color (line 131 to line 134).

Question 7

the authors should precise at what stage of follow up the reported VAS and opening values were registered.

Reply:Thanks for your question. This group of patients was analyzed by visual analog scale (VAS) for pain and improvement of maximal mouth opening comfort before treatment and 6 months after treatment. The corresponding part is on page 2 with red color (line 68 to line 70).

Question 8.

did  the patient undergo post op MRI ? what are the results in terms of stability of disk repositioning ? was the new bone formation reported in image 5 and 6 common in all the patients ? 

it would be very interesting to know the result of post op imaging.

Reply:Thanks for your question. As you know, some patients may complain the expensive MRI or discomfort experience with MRI.  All in all, 10 patients underwent post op MRI totally. The new bone formation was found in 5patients. The corresponding part is on page 7 with red color (line 145 to line 146).

Question 9.

concerning image 5 and  6 they should be anonimyzed : it is possible  to read the name of the patient.

IMAGE 5-6  there is also  problem with the date: the PREOP MRI seems performed in 2019; the post op MRI is performed on 4/2019 according to the date reported in the scan . How is this possible?

 in images5-6 the age reported on the scan for the patient is 13 years old ( born 4/21/2006).  the material and method section report ages of the patient included in the 32-75 years range. How is that possible ?

Reply:Thanks you very much for your kind remind. We changed images 5-6.

Question 10.

DISCUSSION : the overall structure of the discussion should be reformulated . clear references should be made to the literature reported results for TMJOA , according to stages and different techniques.

lines 144-147  should be reformulated . which are the difference inputs op functional recovery if the surgical methods is always the same ?

the conclusions . especially concerning occlusion relationships and reconstruction of internal structure of the joint  are not supported by the data reported in the study  (lines204-210). in  my opinion the paragraph should be reformulated and made more adherent to the data reported in the study.

Reply:Thanks for your question. We agree with you. To better reformulate the discussion, these parts were deleted(lines 144-147, lines 204-210).

We appreciate your work earnestly, and hope that the correction will meet with approval. Once again, thank you very much for your comments and suggestions.

Reviewer 3 Report

Dear Authors,

This study will evaluate the therapeutic effect of chitosan injection in patients with Temporomandibular joint osteoarthrosis (TMJOA) undergoing temporomandibular joint disc reduction and fixation.

The study was in line with the aims of the journal. 

However, there are some issues that should be addressed.

Abstract

Abstract: The abstract should be a total of about 200 words maximum. The abstract should be a single paragraph and should follow the style of structured abstracts, but without headings: 1) Background: Place the question addressed in a broad context and highlight the purpose of the study; 2) Methods: Describe briefly the main methods or treatments applied. Include any relevant preregistration numbers, and species and strains of any animals used. 3) Results: Summarize the article's main findings; and 4) Conclusion: Indicate the main conclusions or interpretations.

The abstract should be an objective representation of the article: it must not contain results, which are not presented and substantiated in the main text and should not exaggerate the main conclusions.

(https://www.mdpi.com/journal/jcm/instructions#figures

Temporomandibular joint osteoarthrosis (TMJOA) is a common clinical disease.” Please remove from the Objective.

Introduction

I suggest improving the introduction section, which is poor.

After definition, please report the classification according to Diagnostic Criteria for TMD (DC/TMD) Axis I. Thus, report that TMD could be divided in muscle disorders (including myofascial pain with and without mouth opening limitation) or intra-articular disorders (including disc displacement with or without reduction and mouth opening limitation, arthralgia, and arthritis). Schiffman E, Ohrbach R, Truelove E, et al. Diagnostic Criteria for Temporomandibular Disorders (DC/TMD) for Clinical and Research Applications: recommendations of the International RDC/TMD Consortium Network* and Orofacial Pain Special Interest Group. J Oral Facial Pain Headache. 2014;28(1):6-27.).

Moreover, report epidemiological data, reporting that temporomandibular disorder is the second most common musculoskeletal disorder that causes pain and disability (cite and refer to: Valesan LF, Da-Cas CD, Réus JC, Denardin ACS, Garanhani RR, Bonotto D, Januzzi E, de Souza BDM. Prevalence of temporomandibular joint disorders: a systematic review and meta-analysis. Clin Oral Investig. 2021 Feb;25(2):441-453. doi: 10.1007/s00784-020-03710-w. Epub 2021 Jan 6. PMID: 33409693. And Jin LJ, Lamster IB, Greenspan JS, Pitts NB, Scully C, Warnakulasuriya S. Global burden of oral diseases: emerging concepts, management and interplay with systemic health. Oral Dis 2016; 22(7):609-19. Doi: 10.1111/odi.12428).

Report the commonly conservative treatments for arthrogenous TMD (cite and refer to: Ferrillo M, Nucci L, Giudice A, Calafiore D, Marotta N, Minervini G, d'Apuzzo F, Ammendolia A, Perillo L, de Sire A. Efficacy of conservative approaches on pain relief in patients with temporomandibular joint disorders: a systematic review with network meta-analysis. Cranio. 2022 Sep 23:1-17. doi: 10.1080/08869634.2022.2126079.) and then report the conditions that allow to surgical interventions. Thus report better the most common surgical procedures.

Materials and Methods

Thirty two TMJOA patients aged 32-75 years (average 51.4)”. Please put these information in the  Result Section.

Please report the study design.

Did you obtain the approval from the Ethical Committee? Patients signed the informed consent?

Please report the VAS definition. 

You should report the Table with main values and the standard deviations and the p value.

Discussion.

Please report the study limitations.

The References should be reported according to the Journal Instructions. (https://www.mdpi.com/journal/jcm/instructions#figures

Author Response

Dear Editors and Reviewers:

RE: Analysis of the curative effect of temporomandibular joint disc release and fixation combined with chitosan injection in the treatment of temporomandibular joint osteoarthrosis (ID: jcm-2071099)

We would like to thank the Journal of clinical medicine for giving us the opportunity to revise our manuscript. Your positive and constructive comments and suggestions are all valuable and helpful for revising and improving our paper.

We provide this document to explain, point by point, the details of our revisions in the manuscript and our responses to your comments as follows.
We are looking forward to hearing from you soon.

Kind Regards,

Yours sincerely,

Qing Zhou

E-mail: cbwqz8060@163.com

Reviewer #3:

Dear Authors,

This study will evaluate the therapeutic effect of chitosan injection in patients with Temporomandibular joint osteoarthrosis (TMJOA) undergoing temporomandibular joint disc reduction and fixation.

The study was in line with the aims of the journal. 

However, there are some issues that should be addressed.

Question 1.

Abstract: The abstract should be a total of about 200 words maximum. The abstract should be a single paragraph and should follow the style of structured abstracts, but without headings: 1) Background: Place the question addressed in a broad context and highlight the purpose of the study; 2) Methods: Describe briefly the main methods or treatments applied. Include any relevant preregistration numbers, and species and strains of any animals used. 3) Results: Summarize the article's main findings; and 4) Conclusion: Indicate the main conclusions or interpretations.

The abstract should be an objective representation of the article: it must not contain results, which are not presented and substantiated in the main text and should not exaggerate the main conclusions.

(https://www.mdpi.com/journal/jcm/instructions#figures) 

“Temporomandibular joint osteoarthrosis (TMJOA) is a common clinical disease.” Please remove from the Objective.

Reply: Thank you for your question. We re-wrote the abstract. Objective: The purpose of this study was to evaluate the efficacy of disc release, fixation and chitosan injection in the treatment of temporomandibular joint osteoarthrosis(TMJ-OA). Methods: From March 2021 to March 2022, 32 patients who underwent unilateral reduction and fixation of temporomandibular joint disc release were selected. All patients were diagnosed with TMJ-OA and were treated with chitosan injection. This group of patients was analyzed by visual analog scale (VAS) for pain and improvement of maximal mouth opening comfort before treatment and 6 months after treatment. A paired t-test was used to evaluate the treatment effect, and P<0.05 indicated that the difference was statistically significant. Results: All 32 patients were successfully treated by surgery and chitosan injection in the second week after operation. The duration of disease in this group ranged from 1 to 10 months, with an average of 5.7 months. After 6 months of follow up, 30 patients were satisfied with the treatment and 2 were unsatisfied. The difference in the treatment effect was found to be statistically significant (P<0.05). Conclusion: Temporomandibular joint disc release and fixation combined with chitosan injection is effective in the treatment of TMJ-OA. The corresponding part is on page 1 with red color (line 7 to line 22).

Question 2.

Introduction

I suggest improving the introduction section, which is poor.

After definition, please report the classification according to Diagnostic Criteria for TMD (DC/TMD) Axis I. Thus, report that TMD could be divided in muscle disorders (including myofascial pain with and without mouth opening limitation) or intra-articular disorders (including disc displacement with or without reduction and mouth opening limitation, arthralgia, and arthritis). Schiffman E, Ohrbach R, Truelove E, et al. Diagnostic Criteria for Temporomandibular Disorders (DC/TMD) for Clinical and Research Applications: recommendations of the International RDC/TMD Consortium Network* and Orofacial Pain Special Interest Group. J Oral Facial Pain Headache. 2014;28(1):6-27.).

Moreover, report epidemiological data, reporting that temporomandibular disorder is the second most common musculoskeletal disorder that causes pain and disability (cite and refer to: Valesan LF, Da-Cas CD, Réus JC, Denardin ACS, Garanhani RR, Bonotto D, Januzzi E, de Souza BDM. Prevalence of temporomandibular joint disorders: a systematic review and meta-analysis. Clin Oral Investig. 2021 Feb;25(2):441-453. doi: 10.1007/s00784-020-03710-w. Epub 2021 Jan 6. PMID: 33409693. And Jin LJ, Lamster IB, Greenspan JS, Pitts NB, Scully C, Warnakulasuriya S. Global burden of oral diseases: emerging concepts, management and interplay with systemic health. Oral Dis 2016; 22(7):609-19. Doi: 10.1111/odi.12428).

Report the commonly conservative treatments for arthrogenous TMD (cite and refer to: Ferrillo M, Nucci L, Giudice A, Calafiore D, Marotta N, Minervini G, d'Apuzzo F, Ammendolia A, Perillo L, de Sire A. Efficacy of conservative approaches on pain relief in patients with temporomandibular joint disorders: a systematic review with network meta-analysis. Cranio. 2022 Sep 23:1-17. doi: 10.1080/08869634.2022.2126079.) and then report the conditions that allow to surgical interventions. Thus report better the most common surgical procedures.

Reply: Thank you for your question. We agree with you. Temporomandibular joint disorders(TMD) could be divided in muscle disorders (including myofascial pain with and without mouth opening limitation) or intra-articular disorders (including disc displacement with or without reduction and mouth opening limitation, arthralgia, and arthritis). TMD is the second most common musculoskeletal disorder that causes pain and disability. Although nonsurgical interventions, such as physical and medical treatments, are more prevalent in the clinic, surgical treatment, including open and arthroscopic procedures, is required in some TMJOA patients with severe symptoms. The corresponding part is on page 1 with red color (line 33 to line 40) and page 4(line 41).

Question 3.

Materials and Methods

“Thirty two TMJOA patients aged 32-75 years (average 51.4)”. Please put these information in the  Result Section.

Reply: Thank you for your question. We removed these information and put them in the result section. Thirty-two TMJ-OA patients aged 32-75 years (average 51.4) The corresponding part is on page 6 with red color (line 129).

Please report the study design.

Reply: Thank you for your question. All patients were asked to receive chitosan injection two weeks later after operation. This group of patients was analyzed by visual analog scale (VAS) for pain and improvement of maximal mouth opening comfort before treatment and 6 months after treatment. A paired t-test was used to evaluate the treatment effect, and P<0.05 indicated that the difference was statistically significant.

The corresponding part is on page 2 with red color (line 67 to line 71).

Did you obtain the approval from the Ethical Committee? Patients signed the informed consent?

Reply: Thank you for your question. All subjects gave their informed consent for inclusion before they participated in the study. The study was conducted in accordance with the Declaration of Helsinki, and the protocol was approved by the Ethics Committee of Stomatological Hospital Affiliated to China Medical University (201914).

Informed Consent Statement: Informed consent was obtained from all subjects involved in the study. The corresponding part is on page 2 with red color (line 47 to line 50).

Please report the VAS definition. 

Reply: Thank you for your question. This method uses a scale of 0 to 10 cm that represents the degree of clinical symptoms: 0 indicates no clinical symptoms, and 10 indicates unbearable clinical symptoms. The corresponding part is on page 6 with red color (line 119 to line 121).

You should report the Table with main values and the standard deviations and the p value.

Reply: Thank you for your question. We did as you required. The corresponding part is on table 1 with two figures followed.

Question 4.

Discussion.

Please report the study limitations.

Reply: Thank you for your question. The study has shortcoming because of no control group. But, we used paired t test to assess the therapeutic effect. As we know, the paired t test, a self-controlled statistical method that can eliminate interference associated with outcome variables, was applied. The study focus its attention on the therapeutic effect of surgery combined with chitosan injection. In our cohort, each patient received the same approach. Given the follow up results, we deem that the approach is effective. The corresponding part is on page 10 with red color (line 216 to line 221).

We appreciate your work earnestly, and hope that the correction will meet with approval. Once again, thank you very much for your comments and suggestions.

Round 2

Reviewer 1 Report

Still, the reply of the authors is not convincing  As  the methodology of the study is weak. As the study is presenting a human trial of using biomaterials for IDTMJ.

Reviewer 2 Report

I noted the modifications you introduced in this study but the absence of a control group is a main problem with this study. 

the captions to the image of the MRI ids too short ,.  think you should indicate with an arrow the new bone formation. new bone formation should not be the only parameter considered in the post op MRI . I think the number of MRI is insufficient : you should design a new study or implement the present one with more cases with a more complete imaging follow up follow up. 

Round 2

Reviewer 2:

Question 1:I noted the modifications you introduced in this study but the absence of a control group is a main problem with this study. 

Reply:Thank you for your comments. The study has shortcoming because of no control group. But, we used paired t test to assess the therapeutic effect. As we know, the paired t test, a self-controlled statistical method that can eliminate interference associated with outcome variables, was applied. The study focus its attention on the therapeutic effect of surgery combined with chitosan injection. In our cohort, each patient received the same approach. Given the follow up results, we deem that the approach is effective. The corresponding part is on page 10 with red color (line 219 to line 224).

Question 2: the captions to the image of the MRI ids too short ,.  think you should indicate with an arrow the new bone formation. new bone formation should not be the only parameter considered in the post op MRI . I think the number of MRI is insufficient : you should design a new study or implement the present one with more cases with a more complete imaging follow up follow up. 

Reply:Thank you for your comments. We added an arrow to indicate the new bone formation and re-wrote the captions.

This group of patients was analyzed by visual analog scale (VAS) for pain and improvement of maximal mouth opening comfort before treatment and 6 months after treatment. The corresponding part is on page 2 with red color (line 68 to line 70).

As you know, some patients may complain the expensive MRI or discomfort experience with MRI.  All in all, 10 patients underwent post-operative MRI totally. The new bone formation was found in 5patients. The corresponding part is on page 7 with red color (line 145 to line 146).

The study focus its attention on the therapeutic effect of surgery combined with chitosan injection. In our cohort, each patient received the same approach. Given the follow up results, we deem that the approach is effective. The corresponding part is on page 10 with red color (line 222 to line 224).

Dear reviewer, most patients complain about limited mouth opening and pain within TMJ region. After treatment, the mouth opens bigger and the pain relieved. Can’t it be proved that the treatment works?  Why does it have to be verified by a large number of postoperative MRIs?  MRI is a good choice to assess the treatment effect, but, VAS and mouth opening degree is also enough.

This article aims to propose a new treatment approach for temporomandibular joint osteoarthritis: temporomandibular joint disc release and fixation combined with chitosan injection. We have carried out the approach clinically and achieved good results. This idea of the treatment strategy is not a fake and we are not copying anyone else. Given the prestige of your publication, we hope the treatment strategy be learned all around the word.  We hope the revised one can meet with your requirement. Many thanks !

Reviewer 3 Report

Authors modified the text according to the suggestions.

I found this work impactful and it fits well with in the scope of this journal.

Author Response

Authors modified the text according to the suggestions.

I found this work impactful and it fits well with in the scope of this journal.

Reply: Thank you very much for your comments!
